# Relationship between Latent Toxoplasmosis and Depression in Clients of a Center for Assisted Reproduction

**DOI:** 10.3390/pathogens10081052

**Published:** 2021-08-19

**Authors:** Jana Hlaváčová, Jaroslav Flegr, Kateřina Fiurašková, Šárka Kaňková

**Affiliations:** 1Department of Philosophy and History of Science, Faculty of Science, Charles University, 128 44 Prague, Czech Republic; flegr@cesnet.cz (J.F.); sarka.kankova@natur.cuni.cz (Š.K.); 2Department of Zoology, Faculty of Science, Charles University, 128 44 Prague, Czech Republic; katerina.fiuraskova@natur.cuni.cz

**Keywords:** *Toxoplasma gondii*, fertility, depression score, Beck Depression Inventory

## Abstract

Latent infection of the globally spread parasite *Toxoplasma gondii* in humans has been associated with changes in personality and behavior. Numerous studies have investigated the effect of toxoplasmosis on depression, but their results are inconsistent. Our study focused on the effect of latent toxoplasmosis on depression in men and women in association with their fertility. In 2016–2018, we recruited clients (677 men and 664 women) of the Center for Assisted Reproduction and asked them to complete a standardized Beck Depression Inventory-II. In women without fertility problems, we found higher depression scores in *Toxoplasma*-positive than in *Toxoplasma*-negative (*p* = 0.010, Cohen’s d = 0.48). *Toxoplasma*-positive infertile men, on the other hand, had lower depression scores than *Toxoplasma*-negative infertile men (*p* ≤ 0.001, Cohen’s d = 0.48). Our results are consistent with the previously described effects of latent toxoplasmosis, which seem to go in opposite directions regarding the effect on personality and behavior of men and women. Our results could be explained by gender-contrasting reactions to chronic stress associated with lifelong infection. This suggests that due to gender differences in the impact of latent toxoplasmosis, future studies ought to perform separate analyses for women and men.

## 1. Introduction

Toxoplasmosis is a disease caused by the intracellular parasite *Toxoplasma gondii* (*T. gondii*). Its prevalence in the human population ranges between 20–80% depending on age, cultural habits, and environmental factors [1]. *T. gondii*’s definitive hosts are felines, from whose small intestine oocysts of the parasite are released into the environment together with feces. Intermediate hosts of the parasite can be any warm-blooded animal, including humans [2]. Possible sources of postnatally acquired human infection include raw or undercooked meat containing parasite cysts, or food and water contaminated with oocysts from feline excrements [1]. In humans, postnatally acquired toxoplasmosis manifests itself in two clinically distinct phases. The acute phase is characterized by a rapid proliferation of tachyzoites in various cells of the host’s body. Symptoms of this phase resemble those of a viral or bacterial infection and in immunocompetent individuals, this phase spontaneously transitions to a latent stage during which bradyzoites slowly multiply in tissue cysts [1]. Tissue cysts can occur in the central nervous system, skeletal and cardiac muscles, lungs, liver, kidneys, or reproductive organs [3,4]. In infected individuals, the infection persists for the rest of their lives.

It has long been known that latent infection caused by this neurotropic pathogen is associated with specific effects on human personality [5,6] and behavior [7,8,9,10]. Several studies have investigated the impact of latent toxoplasmosis on depression, but their results are inconsistent. For instance, Kamal et al. [11] found that *Toxoplasma*-positive psychiatric patients had depression scores (measured using the Beck Depression Inventory, BDI) that were significantly higher than *Toxoplasma*-negative psychiatric patients. A case-control seroprevalence study [12] and a cross-sectional study [13] showed that psychiatric patients suffering from depression have a significantly higher seroprevalence of toxoplasmosis than control subjects without depression, while another study showed associations between toxoplasmosis and depressive symptoms in female veterans [14] and pregnant women [15]. In a representative study of the Finnish population [16], researchers found higher BDI scores in *Toxoplasma*-positive women than in *Toxoplasma*-negative women. 

On the other hand, a cross-sectional study performed on a nonclinical Czech population found that infected men seem to be protected from unipolar depression [17], while the Third US National Health and Nutrition Survey found no association between *T. gondii* seroprevalence and a history of major depression [18]. A meta-analysis of 29 studies showed that toxoplasmosis should not be considered a risk factor for patients with depression [19] and a meta-analysis of 50 studies found no significant association between anti-*Toxoplasma* antibodies and major depression [20]. Similarly, an ecological study found no association between toxoplasmosis and major depression [21]. 

Depression and anxiety are known to be associated with infertility [22]. Infertility is defined as a couple’s inability to become pregnant after twelve or more months of regular unprotected sex [23]. Approximately one-third of infertility cases in couples are attributed primarily to women, one-third to men, and the remaining one-third to the woman-man interaction, of which 20% are unexplained [24]. A meta-analysis showed higher depression and anxiety scores in infertile couples than fertile ones [25] and elevated infertility-related stress has been observed more in women than in men [26]. Infertile women also reported higher levels of depression when compared with infertile men [27].

The inability to naturally conceive is a common problem for many couples. Latent toxoplasmosis appears to be one of the causes of fertility disorders in humans. In a questionnaire study by Kaňková et al. [28], infected women reported that it took them significantly longer to conceive, to become pregnant at an older age, and experienced more fertility problems overall than uninfected women did. *Toxoplasma*-positive women are thus more likely to require artificial insemination than *Toxoplasma*-negative women. In fact, a higher prevalence of toxoplasmosis has been observed in infertile women than in healthy pregnant women [29,30], in infertile couples than in fertile ones [31], and in infertile men than in fertile ones [32]. Moreover, *Toxoplasma*-positive men had a higher level of anti-sperm antibodies than *Toxoplasma*-negative men did [31]. Hlaváčová et al. [33] found a significantly higher incidence of fertility problems in *Toxoplasma*-positive than in *Toxoplasma*-negative men. They also showed that latent toxoplasmosis negatively affects sperm count and motility.

Although the adverse effects of latent toxoplasmosis on human fertility and fertility problems related to depressive symptoms have been repeatedly observed, no study has ever tested the association between latent toxoplasmosis, fertility, and depression. The aim of this study is thus to analyze the effect of latent toxoplasmosis on depression in men and women in relation to their fertility.

## 2. Results

### 2.1. Characteristics of Sample

The final dataset contained 664 women with a mean age of 33.3 years (SD = 4.8), of whom 172 (25.9%) were *Toxoplasma*-positive. The mean age of infected women was higher than the mean age of uninfected women (*p* = 0.027; Table 1). We found no differences in the size of place of residence, level of education, smoking, or prevalence of fertility disorders between *Toxoplasma*-positive and *Toxoplasma*-negative women (see Table 1 for more details on sample characteristics).

The dataset also contained 677 men with mean age of 35.6 years (SD = 5.4), of whom 164 (24.2%) were *Toxoplasma*-positive. The mean age of infected men did not differ from the mean age of uninfected men (*p* = 0.116; Table 1). We found that *Toxoplasma*-positive men were significantly more likely to reside in a place with fewer inhabitants than *Toxoplasma*-negative men were (χ^2^ = 14.1, *p* = 0.015). We found no differences in level of education, smoking, or prevalence of fertility disorders between *Toxoplasma*-positive and *Toxoplasma*-negative men. (For further details of sample characteristics, see Table 1).

The prevalence of toxoplasmosis in women (25.9%) did not significantly differ from the prevalence of toxoplasmosis in men (24.2%, χ^2^ = 0.503, *p* = 0.478). The BDI-II score was significantly higher in women than in men (Tau = −0.081, *p* ≤ 0.001, Cohen’s d = 0.25), in fertile women than fertile men (Tau = −0.072, *p* = 0.016, Cohen’s d = 0.22), and in infertile women than in infertile men (Tau = −0.099, *p* < 0.001, Cohen’s d = 0.32).

### 2.2. A Study of Depression in Women

Partial Kendall correlation controlled for age showed no significant differences in BDI-II score between infected and uninfected women (*p* = 0.494) or between women with and without a diagnosed fertility disorder (*p* = 0.089). In follow-up analyses, we assessed the influence of toxoplasmosis on depression separately for fertile and infertile women. In fertile women, we found a higher BDI-II score in *Toxoplasma*-positive than in *Toxoplasma*-negative women (Tau = 0.145, *p* = 0.010, Cohen’s d = 0.48). In infertile women, we found no significant difference in BDI-II score between *Toxoplasma*-positive and *Toxoplasma*-negative women (*p* = 0.717). For more details of analyses, see Table 2 and Figure 1.

This table shows the results of partial Kendall correlation controlled for age in women and men according to toxoplasmosis status and fertility problems.

### 2.3. A Study of Depression in Men

Partial Kendall correlation controlled for age showed a higher BDI-II score in *Toxoplasma*-negative than in *Toxoplasma*-positive men (Tau = −0.075, *p* = 0.003, Cohen’s d = 0.25). The relationship remained significant even after filtering out the influence of size of residence (Tau = −0.066, *p* = 0.011, Cohen’s d = 0.22). The results showed no difference in BDI-II score between men with and without a diagnosed fertility disorder (*p* = 0.295). In fertile men, we found no significant difference in the BDI-II score between *Toxoplasma*-positive and *Toxoplasma*-negative men (*p* = 0.173). In the group of infertile men, on the other hand, we found a higher BDI-II score in *Toxoplasma*-negative than in *Toxoplasma*-positive men (Tau = −0.152, *p* ≤ 0.001, Cohen’s d = 0.48). See Table 2 and Figure 1 for more details of the analyses.

## 3. Discussion

We studied the effect of latent toxoplasmosis on depression in a specific group of men and women, namely the clients of a fertility clinic. Similarly to Faramarzi et al. [27], who studied the differences in BDI scores in women and men undergoing artificial insemination, we found higher depression levels in women than in men. On the other hand, although a higher prevalence of toxoplasmosis has been repeatedly demonstrated in women than in men in the Czech Republic [34,35], we did not find this difference in our study. This may be due to our atypical sample of participants (clients of the Center for Assisted Reproduction) because a higher prevalence of toxoplasmosis has been observed in infertile men [32,33] and infertile women [30].

We found no significant difference in depression levels between *Toxoplasma*-positive and *Toxoplasma*-negative women in the dataset as a whole; however, in women without fertility disorders we found that *Toxoplasma*-positive women are significantly more depressed than those who are *Toxoplasma*-negative. These results are consistent with studies that have shown higher depression levels in *Toxoplasma*-positive veteran women [14] and in pregnant women [15]. We found no significant difference in depression levels between *Toxoplasma*-positive and *Toxoplasma*-negative women who had been diagnosed with fertility problems. Depression scores in these two groups were similar to those found in *Toxoplasma*-positive women without fertility problems. Infertility in women is associated with increased depression [22] and, indeed, in our sample the negative impact of infertility on depression in women was close to statistical significance (*p* = 0.089). The impact of toxoplasmosis on depression may thus be masked by the stronger effect of infertility. Our sample contained more women with diagnosed fertility issues (68%) than those without and it also contained less *Toxoplasma*-positive (26%) than *Toxoplasma*-negative women, which could explain why we found no significant effect of toxoplasmosis in the sample of women as a whole.

We found a significant difference in depression levels between *Toxoplasma*-positive and *Toxoplasma*-negative men in the whole dataset and in the subset of men with a pathological spermiogram. Consistent with a previously published study [17], our results also indicate that *Toxoplasma*-positive men could be protected from depression. A host’s infection is characterized by elevated levels of IL-10 [36,37,38], which can reduce depression via its immunosuppressive and anti-inflammatory activities [39,40]. Flegr et al. [41] suggest that this could reduce BDI-II depression scores in nonclinical populations of *Toxoplasma*-positive men. This mechanism alone, however, cannot explain why the depression-protective effect of toxoplasmosis was not observed in women and why *Toxoplasma*-positive women without fertility problems had significantly higher depression scores than *Toxoplasma*-negative women. 

Interestingly, our results show that latent toxoplasmosis affects depression levels in the opposite direction in men and women: they increase in women and decrease in men. Significant differences between men and women in the effect of latent toxoplasmosis on personality changes are known to exist. *Toxoplasma*-positive men seem to be less observant of rules and be more suspicious, jealous, and dogmatic than *Toxoplasma*-negative men, while *Toxoplasma*-positive women are more warm-hearted, easygoing, conscientious, persistent, and moralistic than *Toxoplasma*-negative women [5,6,7,42]. These opposite behavioral responses to *T. gondii* infection have been explained by the contrasting reactions of men and women to the chronic stress associated with lifelong infection [8,9]. To cope with stress, women usually seek and provide social support [43,44,45], while men use more individualistic and antisocial strategies [45,46]. Similarly, an evolutionary explanation suggests that the difference between men’s “fight or flight” and women’s “tend and befriend” response to stress stems from women’s need to protect children and maintain social relationships [47]. From a physiological point of view, Kudielka and Kirschbaum observed sex differences in the HPA axis stress responses [48]. It is indeed possible that stress-related mechanisms could play a role in the observed differences in the effect of toxoplasmosis on depression scores in men and women detected in our study. 

Recent meta-analyses which portrayed no relationship between toxoplasmosis and major depression [19,20] were based on samples of psychiatric patients. In the present study, on the other hand, we excluded subjects who were taking antidepressants from our analyses. Moreover, some the studies referenced above examined the relationship between toxoplasmosis and depression based on pooled data collected from both sexes, which would have obscured the above-mentioned differences between the sexes. In the meta-analysis of Nayeri et al. [19], it was impossible to separately analyze men and women because the data were not available in all studies covered by the article. The results of Suvisaari et al. [16], who measured depression using the Beck Depression Inventory (as in our study), support the hypothesis of sex-differential outcomes. They found higher BDI scores in *Toxoplasma*-positive individuals in a representative Finnish sample. When, however, they performed analyses separately for men and women, they found a higher BDI score only in *Toxoplasma*-infected women than *Toxoplasma*-uninfected women. In men, they found no such difference in BDI scores between *Toxoplasma*-infected and uninfected individuals. 

Differences in the results of various studies may also be attributed to differences in the measurement of depression. Some studies did not measure the severity of depression and only examined the prevalence of toxoplasmosis in psychiatric patients compared to healthy controls; this is summarized in a meta-analysis by Sutterland et al. [20]. In our study, we measured depression using a standardized Czech version [49] of the Beck Depression Inventory-II [50]. Although Kamal et al. [11] found significantly higher depression scores measured by the Beck Depression Inventory in *Toxoplasma*-positive psychiatric patients than in *Toxoplasma*-negative patients, unfortunately they did not perform the analysis separately in men and women.

Behavioral changes associated with latent toxoplasmosis have long been studied and interpreted within the theoretical framework of the so-called ‘manipulation hypothesis’, which states that parasites can alter the behavior of their hosts so as to aid their transfer from intermediate hosts to a definitive host by predation [51]. However, association does not necessarily mean causality. The observed changes in behavior and personality between *Toxoplasma*-positive and *Toxoplasma*-negative subjects may be either the cause or the effect of toxoplasmosis. Changes caused by toxoplasmosis could be either the product of *Toxoplasma’s* above-mentioned manipulative activity [51], side effects of pathological processes in the infected organism, or adaptive or maladaptive host responses to parasitic infection. However, it is also possible that individuals with different behaviors and personalities may differ in their susceptibility to *Toxoplasma* infection or exhibit different levels of risk-taking behaviors that lead to infection. In human studies, it is impossible to directly test the direction of causality between these phenomena. Results of longitudinal studies in humans [6,7] and experiments in laboratory animals [52,53,54] do, however, provide support for the hypothesis of infection-induced behavioral changes. 

The likelihood of *T. gondii* infection is known to increase with age. In our dataset, the mean age was higher in *Toxoplasma*-positive women than in *Toxoplasma*-negative ones. In men, we found no association between age and *Toxoplasma* status. A recent epidemiological study [34] conducted in the Czech Republic had shown that the prevalence of toxoplasmosis in boys and girls is similar until the age of 19. At about 30 years of age, the prevalence is significantly higher in women than in men. After this age, the prevalence in men stagnates or decreases, while in women it increases until the age of 50. The traditional explanation for this increasing prevalence of toxoplasmosis in women of childbearing age is their involvement in cooking and tasting raw meat [35]. The possible transmission of *T. gondii* from men to women by sexual intercourse [55,56] and oral sex [57] is also discussed in the literature. It is thus possible that in women, infection rates increase more markedly with age than in men. Women seem to have a greater chance of encountering more sources of *T. gondii* infection than men do. One of the main risk factors for *Toxoplasma* infection is the size of place of residence [58]. In our study, we observed an effect of size of place of residence on toxoplasmosis in men only. This study was part of a larger study on the effects of latent toxoplasmosis on human fertility. It included an epidemiological study [56] which showed that the main risk factors for women were the size of place of residence in childhood and infection of their sexual partner. Other risk factors connected with *T. gondii* infection—such as eating poorly washed root vegetables and raw meat, contact with garden soil, and cat keeping—were not significantly associated with toxoplasmosis in women. In men, however, the authors observed more typical sources of *T. gondii* infection, namely the size of place of residence in childhood and contact with garden soil.

### Limitations

One possible limitation of this study could be the participation of couples who could not naturally conceive. Depression scores may have been affected not only by respondents’ own fertility problem but also by their partners’ fertility status, which would obscure the results of analysis on the effect of fertility on depression. This may be why, although an earlier meta-analysis detected higher depression scores in infertile than in fertile couples [25], our results show no correlation between depression and fertility disorders.

Another limitation of our study was the use of self-reporting questionnaires on depression in the waiting room of the Center for Assisted Reproduction. Although couples were instructed to work on the questionnaire independently, we cannot be sure they followed the instructions.

## 4. Materials and Methods

### 4.1. Study Design and Participants

This cross-sectional study was part of a larger study of the effect of latent toxoplasmosis on human fertility, which took place in June 2016–June 2018 in collaboration with the Center for Assisted Reproduction at the Gynecological and Obstetric Clinic of the First Faculty of Medicine of Charles University and the General University Hospital in Prague. The study included couples who visited the Center for Assisted Reproduction with fertility problems. Informed consent was obtained from all participants. During a routine examination, blood samples were taken for serological testing for toxoplasmosis. Upon entering the study, participants completed a questionnaire that included socio-demographic questions, questions on medications they are using, and a standardized Czech version [49] of the Beck Depression Inventory-II (BDI-II) [50]. Patients who used antidepressants were excluded from the analyses (18 *Toxoplasma*-negative women, 3 *Toxoplasma*-positive women, 9 *Toxoplasma*-negative men, and 3 *Toxoplasma*-positive men). After three weeks, participants were informed about their serological test results for toxoplasmosis.

The study was approved by the Ethics Committee of the General University Hospital in Prague (No. 384/16; 92/17) and the Institutional review board of the Faculty of Science, Charles University (No. 2015/29).

### 4.2. Questionnaire BDI-II 

The Beck Depression Inventory-II (BDI-II) is a self-reporting scale (21 items) aimed at capturing the cognitive, affective, motivational, and physiological symptoms of depression. The items are rated on a 4-point scale ranging from 0 to 3. The overall score may range from 0 to 63, with higher scores indicating a more pronounced presence of depressive symptoms. The BDI-II showed high reliability (Cronbach’s alpha: in women overall = 0.924, in fertile women = 0.931, in infertile women = 0.927, in men overall = 0.911, in fertile men = 0.897, in infertile men = 0.914). Participants who did not complete the BDI-II questionnaire or omitted over one-fifth of questions were excluded from the study (31 men and 29 women). A total of 53 (8%) incomplete BDI-II questionnaires filled by women where one-fifth or fewer responses were missing and 37 (5%) such questionnaires answered by men were used in the study after the missing answers were substituted with the average score of that respondent’s other answers.

### 4.3. Serological Testing for Toxoplasmosis

Toxoplasmosis testing was performed in the National Reference Laboratory for Toxoplasmosis at the National Institute of Public Health in Prague by the complement fixation test and ELISA IgG test (TestLine Clinical Diagnostics). Negative results in both tests indicated that the patient was not infected with *T. gondii*, while positive results indicated the presence of anamnestic anti-*Toxoplasma* antibodies. Participants with inconclusive test results (one test was positive, the other negative) were excluded from further analyses (23 men and 39 women).

### 4.4. Fertility Assessment

Fertility examination was performed at the Center for Assisted Reproduction. We obtained data on diagnosed fertility disorders from a medical database. The disorders diagnosed in women included the tubular factor infertility, anovulation, endometriosis, ovarian failure, or immunological infertility. For the purpose of this study, we identified women with these disorders as infertile. Women who were not diagnosed with any of these disorders (and were marked as “without a pathological finding” in the medical database) were identified as fertile. For 207 women, specific diagnosis was not available in the medical database and these women were not included in fertility analyses.

The men underwent semen examination in the laboratory at the Center for Assisted Reproduction, where spermiograms which indicated lower sperm count, fewer motile sperm, or fewer sperm of normal morphology than the WHO reference limits, were marked as pathological [59]. Men with such spermiograms were then identified for the purpose of this study as infertile. Men who did not have a pathological spermiogram were identified as fertile. A total of 46 men who did not undergo semen examination were not included in further analyses. 

### 4.5. Statistical Analysis

The data were analyzed using Jamovi 1.6.15 [60]. The Mann-Whitney U-test and the χ2 test of association were used to compare the age, size of place of residence, level of education, smoking, and fertility disorders in groups of women and men according to their toxoplasmosis status. The prevalence of toxoplasmosis in men and women was tested by the χ2 test of association. Differences in BDI-II scores in particular groups were tested by a partial Kendall correlation controlled for age. All data are available in the online open-access repository Figshare (doi: 10.6084/m9.figshare.15074175; accessed date: 30 July 2021).

## 5. Conclusions

Our results showed that the effect of toxoplasmosis on depression goes in the opposite direction in men and in fertile women. While toxoplasmosis seems to protect men from depression, it appears to increase the likelihood of depression in women. Our results concur with previous anecdotal observations of a lower incidence of major depression in men with toxoplasmosis [17]. This interaction between toxoplasmosis, sex, and depression could help explain the inconsistent results of previous studies and the large heterogeneity of results reported in meta-analytic studies. The effects of toxoplasmosis on men and women are likely to interfere with each other and the outcome of studies also depends on the male-to-female ratio in the studied sample. Our results suggest that in future studies on the effects of toxoplasmosis on depression in humans, data on men and women should always be analyzed separately. 

## Figures and Tables

**Figure 1 pathogens-10-01052-f001:**
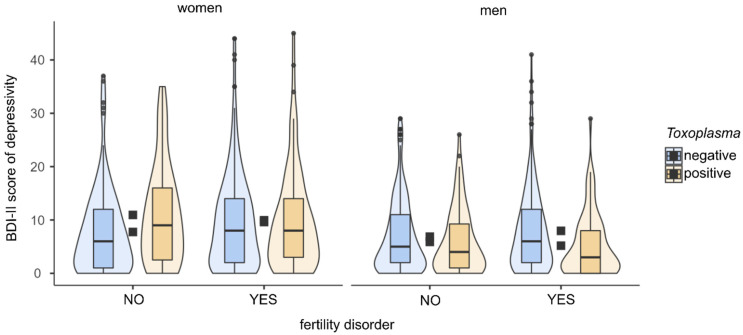
BDI-II scores in women and men according to toxoplasmosis status and fertility problems. The figure shows boxplots with medians, interquartile ranges in violin plots. Black squares show mean depression scores.

**Table 1 pathogens-10-01052-t001:** Characteristics of women and men samples depending on toxoplasmosis.

	Women		Men	
	toxo-neg.	toxo-pos.	toxo-neg.	toxo-pos.
	N = 492	N = 172	N = 513	N = 164
**Mean age** (SD)	33.0 (4.9)	34.0 (4.3)	35.5 (5.4)	36.1 (5.4)
**Size of place of residence** (no. of inhabitants)				
Up to 1000; N (%)	64 (13.3)	27 (16.1)	79 (15.6)	35 (21.6)
1000–5000; N (%)	69 (14.3)	22 (13.1)	59 (11.7)	19 (11.7)
5000–50,000; N (%)	104 (21.6)	36 (21.4)	107 (21.2)	31 (19.1)
50,000–100,000; N (%)	24 (5.0)	7 (4.2)	19 (3.8)	10 (6.2)
100,000–500,000; N (%)	13 (2.7)	1 (0.6)	7 (1.4)	8 (4.9)
Over 500,000; N (%)	208 (43.2)	75 (44.6)	234 (46.3)	59 (36.4)
Missing data	10	4	8	2
**Level of education**				
Highschool without graduation or lower; N (%)	63 (13.0)	26 (15.2)	111 (21.8)	43 (26.9)
Highschool with graduation; N (%)	187 (38.7)	65 (38.0)	196 (38.4)	67 (41.9)
University; N (%)	233 (48.2)	80 (46.8)	203 (39.8)	50 (31.3)
Missing data	9	1	3	4
**Smoking**				
No; N (%)	324 (76.8)	121 (77.6)	301 (70.5)	94 (69.1)
Yes; N (%)	98 (23.2)	35 (22.4)	126 (29.5)	42 (30.9)
Missing data	70	16	86	28
**Fertility disorder**				
No; N (%)	108 (32.0)	38 (31.7)	276 (58.1)	80 (51.3)
Yes; N (%)	229 (68.0)	82 (68.3)	199 (41.9)	76 (48.7)
Missing data	155	52	38	8

**Table 2 pathogens-10-01052-t002:** BDI-II scores in women and men according to toxoplasmosis and fertility problems.

	Women						Men					
	N	Mean	SD	Tau	Cohen’s d	*p*	N	Mean	SD	Tau	Cohen’s d	*p*
Toxo-pos.	172	9.4	9.2	0.018	0.06	0.494	164	5.9	6.4	−0.075	0.25	0.003
Toxo-neg.	492	8.8	8.3				513	7.4	7.4			
Fertile	146	8.6	8.5	0.053	0.16	0.089	356	6.7	6.7	0.028	0.09	0.295
Infertile	311	9.7	9.1				275	7.2	7.3			
Fertile, toxo-pos.	38	10.9	9.4	0.145	0.48	0.010	80	5.9	6.1	−0.48	0.16	0.173
Fertile, toxo-neg.	108	7.8	8.1				276	6.9	6.8			
Infertile, toxo-pos.	82	10.0	9.5	0.014	0.03	0.717	76	5.2	5.9	−0.152	0.48	<0.001
Infertile, toxo-neg.	229	9.6	9.0				199	8.0	7.7			

## Data Availability

The data presented in this study are openly available in FigShare at [doi: 10.6084/m9.figshare.15074175].

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
