# Peer review of "Relationship between Latent Toxoplasmosis and Depression in Clients of a Center for Assisted Reproduction"

_pathogens, 2021, doi:10.3390/pathogens10081052_

Round 1

Reviewer 1 Report

The manuscript of Hlaváčová et al. desribes the study conducted on the effect of latent toxoplasmosis on depression in men and women in relation to their fertility.
In my opinion the article is generally well written and maybe interesting for the epidemiologist as well as parasitologist.

However I have few issues for the authors:

1.In my opinion part from the Introduction (line 42-56) should be shortened or moved to the discussion.

2.Line 300- please use the italics in the name of parasite „ T. gondii”

3.Line 300. I have problem with „subject”. Please consider/change this word for the „patient” or „sample”.

  1. I would suggest that the authors use English editing service for improving instances of awkward syntax etc. It would be valuable for improvement of this paper.

Author Response

1. In my opinion part from the Introduction (line 42-56) should be shortened or moved to the discussion.

Response: The Introduction (line 42-56) has been shortened.

2. Line 300- please use the italics in the name of parasite „T. gondii”

Response: We used the italics in the name of parasite „T. gondii”.

3. Line 300. I have problem with „subject”. Please consider/change this word for the „patient” or „sample”.

Response: We changed the word „subject” to the word „ patient”.

4. I would suggest that the authors use English editing service for improving instances of awkward syntax etc. It would be valuable for improvement of this paper.

Response: The previous version was reviewed only by the Czech language editor (Dr. Pilátová). The current version of the manuscript has been also edited by a native speaker (Lincoln Truesdale Cline). The changes are recorded in track changes.

Reviewer 2 Report

This is a very well written and interesting paper adding to the current data on associations between latent toxoplasmosis infection and depression, investigated in males and females attending a fertility clinic. Of particular interest are the reported gender differences. Introduction and discussion are very comprehensive, and help the reader understand the results in the broader context of current knowledge. I have no suggestions for revision or corrections.

Author Response

This is a very well written and interesting paper adding to the current data on associations between latent toxoplasmosis infection and depression, investigated in males and females attending a fertility clinic. Of particular interest are the reported gender differences. Introduction and discussion are very comprehensive, and help the reader understand the results in the broader context of current knowledge. I have no suggestions for revision or corrections.

Response: Thank you for your review.